# SIGMAR1 Confers Innate Resilience against Neurodegeneration

**DOI:** 10.3390/ijms24097767

**Published:** 2023-04-24

**Authors:** Simon Couly, Yuko Yasui, Tsung-Ping Su

**Affiliations:** Cellular Pathobiology Section, Integrative Neuroscience Research Branch, Intramural Research Program, National Institute on Drug Abuse, NIH, 333 Cassell Drive, Baltimore, MD 21224, USA

**Keywords:** Sigma-1 receptor, neurodegenerative disorders, mitochondria-associated endoplasmic reticulum membrane (MAM)

## Abstract

The sigma-1 receptor (SIGMAR1) is one of a kind: a receptor chaperone protein. This 223 amino acid-long protein is enriched at the mitochondria-associated endoplasmic reticulum membrane (MAM), a specialized microdomain of the endoplasmic reticulum that is structurally and functionally connected to the mitochondria. As a receptor, SIGMAR1 binds a wide spectrum of ligands. Numerous molecules targeting SIGMAR1 are currently in pre-clinical or clinical development. Interestingly, the range of pathologies covered by these studies is broad, especially with regard to neurodegenerative disorders. Upon activation, SIGMAR1 can translocate and interact with other proteins, mostly at the MAM but also in other organelles, which allows SIGMAR1 to affect many cellular functions. During these interactions, SIGMAR1 exhibits chaperone protein behavior by participating in the folding and stabilization of its partner. In this short communication, we will shed light on how SIGMAR1 confers protection against neurodegeneration to the cells of the nervous system and why this ability makes SIGMAR1 a multifunctional therapeutic prospect.

## 1. Introduction

The sigma-1 receptor (SIGMAR1) is a 223 amino acid-long protein that is ubiquitously expressed throughout mammalian tissues and cell types [1,2,3]. At the subcellular level, SIGMAR1 is located in an endoplasmic reticulum (ER) functional subdomain called the mitochondria-associated endoplasmic reticulum membrane (MAM) [4,5]. More than four decades after its discovery and two decades after its sequencing, SIGMAR1 is still not satisfactorily understood [6,7,8,9]. Indeed, several questions remain and are hotly debated, such as what structure, conformation, or orientation this protein has, but also, when SIGMAR1 is endogenously activated and by what endogenous ligands [10,11]. Interestingly, various exogenous ligands are shown to interact with SIGMAR1, and depending on their chemical structure, they differentially affect the protein conformation and activate, inactivate, or potentiate SIGMAR1 functions [4]. Although the endogenous activation of SIGMAR1 is less clear, studies note that calcium depletion at the ER induces an effect similar to that of exogenous agonistic ligation. However, despite these remaining questions, there is broad consensus that SIGMAR1 is able to confer an innate resilience against neurodegeneration in various pathological contexts [12,13,14,15,16,17]. Thus, an important question arises: what explains the amazing versatility of SIGMAR1? The answer lies in how this ER-resident protein impacts several key mechanisms that are involved in neurodegenerative disorder physiopathology.

## 2. The Relevance of the Unfolded Protein Response (UPR) in Neurodegenerative Contexts

The ER is heavily involved in post-translational modifications and protein folding, but also in the clearance of misfolded proteins. The mechanism deployed by the ER to regulate unfolded proteins is called the unfolded protein response (UPR) [18]. In mammalian cells, the UPR is governed by three signaling pathways involving IRE1 (inositol-requiring transmembrane kinase/endoribonuclease 1α), ATF6 (activating transcription factor 6), and PERK (PKR-like ER kinase) [19]. Eventually, the activation of these pathways leads to changes in the expression of genes that mediate protein conformational changes or degradation of the unfolded protein. Proteins that facilitate the folding of other proteins are called “chaperone proteins”; SIGMAR1 is one such example. Dysregulation of UPR or the over-accumulation of misfolded proteins leads to “ER stress”, which eventually triggers the apoptosis pathway. ER stress is found in the most prominent neurodegenerative disorders such as Alzheimer’s disease (AD), Parkinson’s disease (PD), Amyotrophic Lateral Sclerosis (ALS), Huntington’s disease (HD), and prion diseases [20,21]. Although the causes of ER stress and unfolded protein accumulation may differ among neurodegenerative disorders, pharmacological targets that enhance the UPR are known to delay or reverse those pathological conditions [19,20].

Mori et al. observed that SIGMAR1, upon ligand activation, can translocate along the ER membrane to interact with and facilitate the IRE1 pathway without impacting the ATF6 or PERK pathways [22]. SIGMAR1 binds and stabilizes IRE1 during ER stress, which is induced either by an ER calcium depletion or an increase in unfolded proteins. Furthermore, Mori et al. noted that this stabilization is necessary for the proper and functional dimerization and phosphorylation of IRE1 during ER stress. The downstream signaling in the IRE1 pathway, especially the splicing of mRNA *xbp1*, is reduced via SIGMAR1 deletion during ER stress. The E102Q mutation on SIGMAR1, which is known to be responsible for a rare form of ALS, upregulates ER stress in the overexpressed cell lines (NSC34 and MCF-7) and E102Q patients’ lymphoblastoid cells [23]. It is also important to note that there is a UPR specific to mitochondria: UPRm [24]. This response seems to be enhanced in transgenic mice with overexpressed 4E-BP1, which is part of the mTOR pathway and therefore plays a fundamental role in the control of pro-survival gene transcription [25]. Wang et al. observed that the knock-out of SIGMAR1 in human embryonic kidney 293T cells decreased the expression of 4E-BP1, which suggests that SIGMAR1 does not only affect UPR from the ER but also from the mitochondria, since 4E-BP1 can impact UPRm [26]. Finally, Mori et al. observed that, like SIGMAR1, IRE1 is enriched at the MAM in Chinese hamster ovarian cells. This sub-localization of IRE1 allows the ER stress sensor to react properly to mitochondria-derived reactive oxygen species (ROS) [22].

## 3. Mitochondria-Associated ER Membrane (MAM): Critical Interface between Two Fundamental Organelles

The MAM is a region of the ER that structurally and functionally connects the ER to the mitochondria. In mammalian cells, including nervous system cells, the MAM fulfills crucial functions such as the regulation of mitochondrial morphology and activity, lipid metabolism, calcium homeostasis, autophagy, and mitophagy [27,28,29,30]. Interestingly, the proper functions of this interface are dependent on its structure [31]. Although the structure and protein content of MAMs are not yet fully understood, the correlation between the integrity of this ER subregion and neurodegenerative disorders has been observed and well studied [31,32,33]. The MAM occupies the space between the outer mitochondrial membrane and the ER. It is a region smaller than 30 nm and covers 5% to 20% of the mitochondrial surface membrane. This interorganelle space is maintained by tethering proteins such as Mitofusin 2 [34,35] or by protein complexes formed by the vesicle-associated membrane protein-associated protein B and the tyrosine phosphatase interacting protein-51 [36,37]. Another protein complex specific to the MAM is the complex formed by the inositol 1,4,5-trisphosphate receptor type 3 (IP3R3) in the ER side and voltage-dependent anion channel in the mitochondrial side, which are connected by the glucose-regulated protein 75. This complex forms a bridge between the ER and mitochondria, and it is particularly important for the facilitation of ER Ca^2+^ exports to the mitochondria [38].

SIGMAR1 is fundamental to the proper interaction between ER and mitochondria, both functionally and structurally [4,39,40,41,42]. In 2007, Hayashi and Su observed that SIGMAR1 stabilizes IP3R3 at the MAM and thus facilitates the efflux of calcium from the ER to the mitochondria [4]. These results were later corroborated by Bernard-Marissal et al., who demonstrated that having either pharmacologic or genetic inhibition of SIGMAR1 reduced MAM, thereby changing the morphology and the transportation of mitochondria inside the cells [40]. Interestingly, and more recently, the relevance of SIGMAR1 to the MAM was also observed outside the nervous system; Tagashira et al. observed the SIGMAR1-mediated synergy of the ER and mitochondria in cardiomyocytes [42]. The impact of MAM integrity in neurodegenerative disorders is well studied and reviewed by Paillusson et al. [31]. AD, PD, and ALS are all especially concerned with MAM disruption, more specifically the reduction in contact between the ER and mitochondria [43,44,45]. Moreover, the rescue of MAM integrity seems to counteract pathological phenotypes observed in these contexts. For example, Calì et al. observed that α-synuclein, a protein involved in PD, increases the co-localization between mitochondria and endoplasmic reticulum markers in Hela cells [44], while Stoica et al. showed in an ALS model that the overexpression of TDP-43 (wild-type or mutant) in NSC34 cells decreases the mitochondrial surface closely opposed to ER when observed with electron microscopy [43].

The impact of SIGMAR1 on MAM stability also seems to depend on the interaction between SIGMAR1 and cholesterol. This interaction appears to be necessary for the stabilization of microdomains comparable to the plasma membrane’s lipid rafts, and also necessary for ER metabolism [39]. The versatile effects of SIGMAR1 agonists may be explained by the effect of SIGMAR1-containing microdomains on the rapid release of ER-matured proteins [46].

## 4. SIGMAR1 and the Control of Oxidative Stress

The integrity of MAM is necessary for both the ER and mitochondria to function properly. Mitochondrial activity is a double-edged sword because while it produces fundamental molecules necessary for the cellular mechanisms, such as ATP, it also produces reactive oxygen species (ROS), which in high concentrations are harmful to cells. Thus, the fine tuning of mitochondrial metabolism is critical to maintaining physiological conditions. Indeed, one of the most toxic cellular-level phenomena occurring in neurodegenerative disorders is the upregulation of oxidative stress. Most neurogenerative disorders present an abnormal concentration of ROS, so the regulation of ROS is a promising therapeutic target. SIGMAR1 has the potential to be an entry point for the control of ROS concentrations. Goguadze et al. demonstrated that the activation of SIGMAR1 under physiological conditions increases the ROS concentration; however, the same activation of SIGMAR1 in the AD context decreases the enhanced ROS concentration induced by the presence of Aβ1-42 [47]. Su et al. noted that the activation of SIGMAR1 using dehydroepiandrosterone (DHEA) can counteract enhanced ROS concentrations through the activation of Zinc finger protein 179 (Znf179). This also leads to an increased concentration of peroxiredoxin 3 and superoxide dismutase 2, antioxidant enzymes that reduce ROS levels in H_2_O_2_-stimulated mouse neuroblastoma cells [48].

Interestingly, oxidative stress increases the concentration of unfolded proteins, thereby contributing to heightened levels of UPR and increasing ER stress [20]. The ameliorating effect of SIGMAR1 on oxidative stress in neurodegenerative contexts therefore indirectly reduces ER stress.

SIGMAR1 is positioned at the junction of two fundamental organelles affected in most, if not all, neurodegenerative disorders: the mitochondria and ER. Therefore, it is a strong therapeutic target. However, the therapeutic relevance of SIGMAR1 reaches even beyond the MAM.

## 5. Beyond the MAM: Other Entry Points for SIGMAR1 to Counteract Neurodegenerative Disorders

### 5.1. BDNF/TrkB

The brain-derived neurotrophic factor (BDNF) acts as a neurotrophic factor through its interaction with and activation of tropomyosin receptor kinase B (TrkB) [49]. TrkB activation has a pro-survival effect via activation of the phosphoinositide 3-kinase (PI3K) or extracellular signal-regulated kinase (ERK) signaling pathways. However, it should be noted that p75, another target of BDNF, has the opposite effect and facilitates apoptosis. The fine-tuning of BDNF expression, its receptors, and its pathways are therefore additional promising therapeutic targets [50]. The production of functional BDNF involves post-translational modification of three different forms of BDNF: pre-pro-BDNF, pro-BDNF, and mature-BDNF (14kDa). In 2008, Kikuchi-Utsumi and Nakaki found that the chronic injection of the SIGMAR1 agonist cutamesine (SA4503) increased the concentration of mature BDNF in rat models [51]. This opens the possibility for more studies on how SIGMAR1 impacts BDNF. Fujimoto et al. later shed light on the mechanism by which SIGMAR1 enhances the BDNF concentration [52]. They concluded that SIGMAR1 can increase BDNF protein maturation by facilitating post-translational modifications, increasing its secretion. These results are in accordance with observations made by Mysona et al. that highlight an increase in mature-BDNF in mice retina and hippocampus induced by a SIGMAR1 agonist ((+)-pentazocine) treatment and a decreased mature-BDNF in retina from SIGMAR1 knock-out mice [53]. More recently, several studies have observed that pridopidine, a highly selective SIGMAR1 agonist, can rescue the impaired transport of BDNF in ALS and HD [54,55]. Lenoir, Lahaye et al. further described pridopidine as being capable of not only increasing BDNF transport and secretion but also of increasing TrkB-containing vesicles and phosphorylated ERK at the post-synaptic level. These results correlate with those of Francardo et al., who demonstrated that treatment by PRE-084 (SIGMAR1 agonist) in a PD pharmacological mouse model rescued pathological behavior and decreased PD biomarkers [56]. More interestingly, Francardo et al. noted that chronic PRE-084 treatment activates the BDNF/TrkB pathway. They also found that knocking out SIGMAR1 counteracts the positive effect of PRE-084 on PD phenotypes. By enhancing the neuroprotective BDNF/TrkB pathway, SIGMAR1 could be therapeutically utilized to enhance the BDNF/TrkB pathway in disease contexts. For example, it has been shown that SIGMAR1 is able to counteract the physiopathology mechanisms in AD via the BDNF/TrkB pathway [57], and similarly in PD [58], ALS [59], and HD [60].

### 5.2. Voltage-Dependent Potassium Channel

SIGMAR1 can also interact with plasma membrane proteins. Among others, Kourrich et al. found a protein–protein association between SIGMAR1 and the voltage-dependent potassium channel Kv1.2 in the nucleus accumbens tissue. This association increases the trafficking and upregulation of Kv1.2 at the plasma membrane and eventually results in hypoactivity of the medium spiny neurons. Several neurodegenerative disorders, such as HD, involve neuronal hyperexcitability; however, further experiments are necessary to determine whether this action between SIGMAR1 on Kv1.2 is neuroprotective [61].

### 5.3. Neuron’s Morphology

The downstream effects of SIGMAR1′s regulation of oxidative species affect other neuronal metabolic processes, such as dendrite spine formation [62]. Dendrites are not the only neurites that could potentially be impacted by SIGMAR1 [63,64]. For example, SIGMAR1 also interacts with ankyrin-B. The ankyrin family serves as anchor proteins to link membrane proteins to the underlying cytoskeleton. Researchers have also discovered a significant correlation between SIGMAR1 expression and growth cone location, suggesting that SIGMAR1 plays a role in growth cone formation. Takebayashi et al. demonstrated that nerve growth factor (NGF)-induced neurite sprouting is potentiated by SIGMAR1, i.e., the NGF effect increases with the co-incubation of a SIGMAR1 agonist: (+)- pentazocine [63]. It must also be noted that NGF, in the same study, increased SIGMAR1. Considering the fact that SIGMAR1 is particularly present near growth cones, that authors have thus concluded that endogenous SIGMAR1 supports the action of NGF on axon formation. More recently, Mysona et al. observed that (+)-pentazocine treatment reduces axonal loss in optics of a glaucoma model inducing optic nerve degeneration; however, further experiments are needed to be completed to understand the mechanism behind this morphological effect [65].

### 5.4. Tau Phosphorylation

Another hypothesis about the broad neuroprotective effect of SIGMAR1 in various neuropathological contexts was brought by Tsai et al. [66]. In their study, they observed that SIGMAR1 regulates tau phosphorylation and facilitates axonal extension upon myristic acid activation and the enhanced degradation of p35. However, further experiments with prototypic SIGMAR1 ligands must be performed to know whether those effects are specific to activation via myristic acid or whether the effects are present with exogenous ligands.

### 5.5. Nuclear Pore Complex

The nuclear pore is another key element of neurodegenerative disorders in which SIGMAR1 is involved [67]. Lee et al. demonstrated using cellular models that SIGMAR1 is present at the nuclear pore complex, the domain of the nuclear membrane that allows the transport of macromolecules between the cytosol and nucleus [68]. At the nuclear pore complex, SIGMAR1 stabilizes nucleoporins, the proteins that constitute the nuclear pore complex, suggesting that SIGMAR1 participates in transport between the cytosol and nucleus. In the same research article, authors observed that SIGMAR1 can bind with toxic RNA and clear it from the cytosol (e.g., *G4C2-RNA*, known to induce ALS). Furthermore, SIGMAR1 interacts with the nucleoporin POM121, which is involved in the transport of the transcription factor EB (TFEB), a fundamental player in autophagy [69]. The impact of SIGMAR1 on the nuclear pore complex and the clearance of toxic RNA may be translatable to neurodegenerative disorders beyond ALS. Recent studies give weight to the hypothesis that the role of SIGMAR1 at the nuclear pore complex could be relevant to counteract the physiopathology of various neurodegenerative disorders [70].

## 6. SIGMAR1 and Glial Cells

SIGMAR1 also acts on intracellular calcium concentrations in microglia, inducing microglia-specific effects [71]. Hall et al. noted that SIGMAR1 activation inhibits the microglial inflammatory response induced by the application of lipopolysaccharide. These observations corroborate the results from Francardo et al., which show that PRE-084 treatment of a PD mouse model reduces microglia activation [56]. The neuroprotective role of SIGMAR1 involves more non-neuronal cells than just microglia. Zhao et al. demonstrated that the deletion of SIGMAR1 from optic nerve head astrocytes inhibits co-cultured retinal ganglion cells’ neurite growth and increases cell death; therefore, SIGMAR1 must be critical for the supportive function of astrocytes on neurons [72]. This hypothesis was confirmed in a later study from the same authors, which showed that (+)-pentazocine application after oxygen/glucose deprivation increased astrocyte reactivity [73]. Nevertheless, it seems that the knocking-out of SIGMAR1 in mice increases the expressions of GFAP (glial fibrillary acidic protein), astrocyte marker, and Nrf2 (nuclear factor erythroid 2-related factor 2; a key transcription factor in antioxidant mechanism) [74]. Interestingly, in the same study, Weng et al. observed no interaction between Nrf2 and SIGMAR1. Several other studies do demonstrate the enhancement of the Nrf2 pathway following a pharmacological activation of SIGMAR1 [75,76]. Contrary to the results of Weng et al., the findings from Barwick et al. showed a direct interaction between SIGMAR1 and Nrf2 [76]. It is often assumed that the deletion of SIGMAR1 would have the same effects as the pharmacological inactivation SIGMAR1 (and opposite effects of SIGMAR1 activation); however, the results described before demonstrate that both SIGMAR1 deletion and activation induce the Nrf2 pathway. This could be explained by the differences between the models, but also perhaps by the compensatory mechanisms on cellular metabolism triggered by the constitutive deletion of SIGMAR1. These compensatory mechanisms may possibly activate the Nrf2 pathway to balance the increased oxidative stress inherent for cells without SIGMAR1.

Liu et al. noted that SIGMAR1 activation in astrocytes supporting the blood–brain barrier (BBB) can also increase glia-derived neurotrophic factor (GDNF) secretion, which in turn enhances the integrity of the BBB [70]. BBB disruption has been well characterized in several neurodegenerative disorders [77]. Another glial cell type whose specific function is affected by SIGMAR1 is oligodendrocytes [78]. It has been suggested that lipid rafts have a crucial role in oligodendrocyte survival [79,80]. Given the similarities between the ER lipid rafts mentioned earlier in Section 3 and the myelin sheaths of oligodendrocytes, Hayashi and Su examined whether those myelin sheaths may also have an association with SIGMAR1. They accordingly observed high expressions of SIGMAR1 clusters present in the myelin sheaths of oligodendrocytes and concluded that SIGMAR1 in oligodendrocytes is necessary for the formation of proper myelin sheaths and the maturation of oligodendrocytes.

Altogether, these results show that SIGMAR1 is a versatile target due to its involvement in many key mechanisms involved in prominent neurodegenerative disorders. These mechanisms include ER stress, UPR, and mitochondrial activity, but also several functions that occur more independently from the MAM, such as the BDNF/TrkB signaling pathway and nuclear transport (Figure 1). Millions of people suffer from neurodegenerative disorders worldwide, yet there is no effective treatment to reverse these conditions. Further studies on SIGMAR1 and its ligands will provide crucial knowledge on how to counteract the pathobiological mechanisms underlying neurodegeneration.

## Figures and Tables

**Figure 1 ijms-24-07767-f001:**
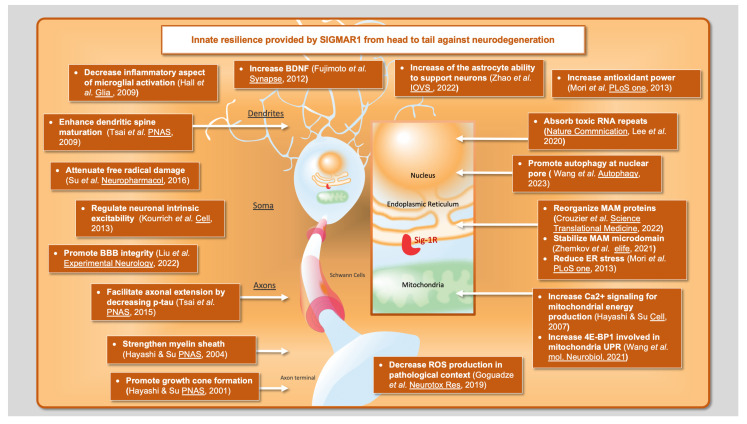
Innate resilience provided by SIGMAR1 from head to tail against neurodegeneration [4,22,26,39,47,48,52,61,62,64,66,68,69,70,71,72,75,78].

## Data Availability

Not applicable.

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
