# Peer review of "SIGMAR1 Confers Innate Resilience against Neurodegeneration"

_ijms, 2023, doi:10.3390/ijms24097767_

Round 1

Reviewer 1 Report

The “SIGMAR1 Confers Innate Resilience Against Neurodegeneration” by Simon Couly and colleagues discussed the role of  SIGMAR1 receptor in the progression of neurodegeneration diseases. In my opinion, the manuscript shows the crucial connections and pathomechanisms, in which receptor is  involved or probably is taking a part. I do not see any crucial issues needed to be farther explained or developed. Nevertheless, I would like to kindly ask for some changes.

From time to time, I can see the description of experimental proves showing particular mechanisms. But, I cannot find the model, on which these findings have been proven. E.g. lines 43-51 or 56-68 ( I do see the type of mutations, but I cannot say whether those come from animal, bacterial or cellular models).
Otherwise, I do not have any further comments.

Author Response

We would like to thank the editor and the reviewers for contributing their time and expertise to improve this manuscript. We have carefully read and integrated the comments and suggestions brought by all four reviewers. We hope this updated version of “SIGMAR1 Confers Innate Resilience Against Neurodegeneration” addresses all concerns and reaches the high publication standard for the International Journal of Molecular Sciences.

Thank you to the reviewer for his/her valuable input. Indeed, model types vary from one publication to another. We have added more precise language about the models used, thereby clarifying the experimental context. As recommended by the editor all revisions are marked up using the “Track Changes” function of word. Here are also their locations:

 Line 47:  “In mammalian cells”

Line 63: “upregulates ER stress in the overexpressed cell lines (NSC34 and MCF-7) and E102Q patients’ lymphoblastoid cells [23]” 

Line 65: “This response seems to be enhanced in transgenic mice with overexpressed 4E-BP1,”

Line 66-67: “in human embryonic kidney 293T cells”

Line 69: “in Chinese hamster ovarian cells”

Line 315: ”mouse neuroblastoma cells”

Line 474: “in the nucleus accumbens tissue.”

Line 476: “of the medium spiny neurons”

Line 501:  “using cellular models”

Reviewer 2 Report

The manuscript deals with current data on key molecular mechanisms that may explain the versatility of the sigma1 receptor in various neurodegenerative pathologies.

It contains an original approach to extrapolate the chaperone properties of sigma1 receptor to protect neurons under cellular stress. Endoplasmic reticulum   stress response (the unfolded protein response) is proposed as a key phenomenon for the chaperone functionality of sigma 1 receptor. Data are presented on the selective ligand-induced interaction of the sigma1 receptor with one of the known molecular signals for triggering endoplasmic stress.

The study lays out  relevant and interesting cellular and molecular aspects in relation to which one can assume a neuroprotective effect of the sigma1 receptor, its participation in the control of the survival-apoptosis balance. First of all, this concerns the molecular mechanisms of interaction between the endoplasmic reticulum and mitochondria, the regulation of transport between the cytoplasm and the nucleus, neuronal morphology, and participation in the survival of gliocytes.

The strength of the study is the selection of the most promising new experimental directions in understanding the mechanisms of the natural resistance of nervous tissue to cellular stress.

The list of cited literature contains approximately 50% of references from the last 5 years.

The manuscript is well structured and easy to understand. It will be useful for a wide range of specialists from molecular biologists to pharmacologists.

Author Response

We thank the reviewer for his/her supportive comments.

Reviewer 3 Report

SIGMAR1 is a protein that is ubiquitously expressed in the mammalian tissues and different cell types. Different ligands bind and affect protein conformation and potentiate SIGMAR1 functions. How this protein is able to confer innate resilience against neurodegeneration in various pathological contexts remain unclear. This happens probably due to its effect on several key mechanisms involved in neurodegenerative disorder pathophysiology. Millions of people suffer from these disorders worldwide. In spite of this, there is no effective treatment to reverse these conditions. SIGMAR1 is a promising target that could provide crucial insights into the underlying mechanisms of these disorders and enable the development of potent therapeutics. In this review/communication, the authors have compiled the existing literature on SIGMAR1. I feel the manuscript is well-written. English seems good and does not require any major editing. I only have a couple of minor requirements.

Minor:

1)    Please explain – border of the nervous system – line 76.

2)    Align all the headings and sub headings.

Author Response

Thank you to the reviewer for his/her positive opinion about our manuscript.

We have made the following adjustments to address the reviewer’s valuable recommendations. As recommended by the editor all revisions are marked up using the “Track Changes” function of word in the updated manuscript.

  • Line 289 : “Beyond the border of the nervous system” was changed to “outside the nervous system” and refers to the impact of SIGMAR1 on the MAM in cells that are not part of the nervous system, g. cardiomyocytes.
  • Thank you, the headings and subheadings have been aligned.

Reviewer 4 Report

Revision Manuscript ijms-2303463

Comments to the Authors:

The short Communication paper from Simon Couly and colleagues entitled “SIGMAR1 Confers Innate Resilience Against Neurodegeneration” focuses on the role of SIGMA1 receptor on the contacts between mitochondria and the endoplasmic reticulum and its possible function in the context of neurodegenerative pathophysiology. Overall, I believe it provides a nice state of the art of SIGMA1 receptor as neuroprotective player. There are a few details that need to be specified, to improve the manuscript.

1.      I suggest also to explain how mams are structurally involved in CNS, to better understand the etiopathology of neurodegeneration alongside mams plasticity, in chapter 3.

2.      MAMs are characterized by several structural parameters, that is, the distance among the 2 parallel membranes and the extension of the contact per se. These 2 different parameters can vary independently from each other: how would this affect the etiopathology of neurodegeneration?

3.      To better understand the role of SIGMA1R on glial cells, I suggest to briefly introduce the SIGMA1R role in lipid metabolism/homeostasis in the context of glial cells.

MINOR POINTS

1.      Line 66: it is not clear, please re-write it.

2.      Line 73-74: Add reference.

3.      Line 83 and 146: specify the agonist of SIGMAR1.

Author Response

We would like to thank the editor and the reviewers for contributing their time and expertise to improve this manuscript. We have carefully read and integrated the comments and suggestions brought by all four reviewers. We hope this updated version of “SIGMAR1 Confers Innate Resilience Against Neurodegeneration” addresses all concerns and reaches the high publication standard for the International Journal of Molecular Sciences.

We think that the suggestions brought by the reviewer are very pertinent. Below is the description of the changes we applied to our manuscript to follow these recommendations.

As recommended by the editor all revisions are also marked up using the “Track Changes” function of word in the updated manuscript.

To answer the point 1 and 2 about the role of MAM structure in the etiopathology of neurodegeneration we added the following paragraph at the beginning of the chapter 3:

Line 74-283 :

 “MAMs are a region of the ER that structurally and functionally connects the ER to the mitochondria. In mammalian cells, including nervous system cells, MAM fulfills crucial functions such as regulation of mitochondria morphology and activity, lipid metabolism, calcium homeostatis, autophagy and mitophagy [27-30]. Interestingly the proper functions of this interface is dependent on the structure [31].Although the structure and protein content of MAM are not yet fully understood, the correlation between the integrity of this ER subregion and neurodegenerative disorders has been observed and well-studied [31-33]{Paillusson, 2016 #22;Yang, 2020 #78;Rossi, 2019 #56}. The MAM occupies the space between the outer mitochondrial membrane and the ER. It is a region smaller than 30 nm and covers 5% to 20% of the mitochondrial surface membrane. This interorganelle space is maintained by tethering proteins like Mitufusin 2 [34, 35] or by protein complexes formed by vesicle-associated membrane protein-associated protein B and tyrosine phosphatase interacting protein-51 [36, 37]. Another protein complex specific to the MAM is the complex formed by inositol 1,4,5-trisphosphate receptor type 3 (IP3R3) in the ER side and voltage-dependent anion channel in the mitochondria side, connected by glucose-regulated protein 75. This complex forms a bridge between the ER and mitochondria and is particularly important for the facilitation of ER Ca2+ export to the mitochondria [38].”

And

Line 291-297 :

“AD, PD and ALS are all especially concerned with MAM disruption, more specifically the reduction of contact between the ER and mitochondria [43-45]. Moreover, the rescue of MAM integrity seems to counteract pathological phenotypes observed in these contexts. For example, Calì et. al. observed that α-synuclein, a protein involved in PD, increases the co-localization between mitochondria and endoplasmic reticulum markers in Hela cells [44], while Stoica et. al. showed in an ALS model that overexpression of TDP-43 (wild-type or mutant) in NSC34 cells decreases mitochondrial surface closely apposed to ER when observed with electron microscopy [43].“

Point 3 :

Line 544-545 : In order to highlight the impact of SIGMAR1 on the glial cells we have added references (line 225) to emphasize the impact of lipid rafts in oligodendrocytes

Minor Points:

1- Thank you for the suggestion. To improve clarity, we split the one sentence into two.

Now line 66 “Wang et al. observe that knock-out of SIGMAR1 decreases the expression of 4E-BP1 in immortalized human embryonic kidney cells. As 4E-BP1 seems to be impacting UPRm this last result suggests that SIGMAR1 does not only affect ER UPR but also UPRm [26].”

2- Thank you, the references were added. (See lines 286 and 288)

3- Line 300: is referring to Sigma-1 Receptor agonists. The confusion comes from our mistake. We corrected “agonist” to “agonists. In the Zhemkov et al. publication they used (+)-SKF-10047.

Line 485: Thank you, we corrected to: “of a SIGMAR1 agonist: (+)- pentazocine”.

Round 2

Reviewer 4 Report

Now the manuscript is suitable for the publication